# Evaluation of Feeding Beta-Hydroxy-Beta-Methylbutyrate (HMB) to Mouse Dams during Gestation on Birth Weight and Growth Variation of Offspring

**DOI:** 10.3390/ani13203227

**Published:** 2023-10-15

**Authors:** Anna S. Clarke, Chris Faulk, Gerald C. Shurson, Daniel D. Gallaher, Lee J. Johnston

**Affiliations:** 1BioMatrix International, Princeton, MN 55371, USA; aclarke0827@gmail.com; 2Department of Animal Science, University of Minnesota, St. Paul, MN 55108, USA; cfaulk@umn.edu (C.F.); shurs001@umn.edu (G.C.S.); 3Department of Food Science and Nutrition, University of Minnesota, St. Paul, MN 55108, USA; dgallahe@umn.edu; 4West Central Research and Outreach Center, University of Minnesota, Morris, MN 56267, USA

**Keywords:** beta-hydroxy-beta-methylbutyrate, birth weight variation, growth performance, mouse, placental efficiency

## Abstract

**Simple Summary:**

In litter-bearing species, fetal competition during pregnancy can decrease the availability of nutrients to some offspring, which can increase the prevalence of low body weight at birth and variability of birth weights. Low birth weight can increase the chance of death early in life and compromise lifetime growth and health. This study was designed to evaluate the efficacy of a nutritional intervention, β-hydroxy-β-methylbutyrate (HMB), to improve uniformity of birth weight in mice. Mice received HMB at low and high doses and during a critical time during pregnancy. Dietary HMB did not influence average birth weight, uniformity of birth weight, growth performance, or body composition of mouse pups born to dams fed HMB during pregnancy. In our mouse model, dietary HMB fed to pregnant mice did not reduce prevalence of low-birth-weight offspring.

**Abstract:**

This study was designed to determine if feeding β-hydroxy-β-methylbutyrate (HMB) to pregnant mice would improve birth weight uniformity and growth performance of offspring. Dams (Agouti A^vy^) were assigned to one of four treatments: control (CON; *n* = 13), low-level HMB (LL; 3.5 mg/g; *n* = 14), high-level HMB (HL; 35 mg/g; *n* = 15), and low-level pulse dose fed from gestational days 6 to 10 (PUL; 3.5 mg/g; *n* = 14). Randomly selected dams (*n* = 27) were euthanized on gestational day 18 to collect placentae and pup weights. The remaining dams gave birth and lactated for 28 days. Dams only received HMB during gestation. Dietary HMB did not influence the performance of dams. Dietary treatment during gestation did not affect litter size or birth weight of pups. Variation was not different among treatments in terms of birth weight of offspring. Placental weights were not affected by treatments. Overall, growth performance of offspring after weaning was similar among all treatments. Body composition of offspring at 5 and 8 weeks of age was similar regardless of HMB treatment during gestation. In conclusion, dietary HMB supplementation in pregnant mice did not affect birth weight, variations in birth weight, or growth performance of offspring.

## 1. Introduction

In litter-bearing species, as litter size increases, fetal competition in utero can cause a decrease in available nutrients for each fetus. This competition can cause suboptimal fetal development and, consequently, reduce individual birth weights and increase within-litter birth weight variation [1]. Fetal growth depends on placental growth and efficiency, which are directly related to the functional efficiency of delivering nutrients and oxygen to fetuses [2]. Placental efficiency is influenced by the weight or size of placentae, the contact surface area with maternal endometrium, placental blood flow, and the ability of the placenta to transfer nutrients to fetuses [3]. Fetal growth also requires the accretion of protein, which has led researchers to evaluate the use of amino acids or their metabolites as potential dietary interventions to reduce the incidence of intrauterine growth retardation (IUGR) and low birth weight in litter-bearing animals. 

Leucine is an essential amino acid, and its metabolite, β-hydroxy-β-methyl butyrate (HMB), may potentially reduce the incidence of IUGR and low-birth-weight neonates. In both in vitro and in vivo studies with rats, leucine stimulated muscle synthesis [4,5] and inhibited proteolysis in muscle in vitro [6]. Similarly, HMB also enhanced skeletal muscle development and reduced proteolysis [7,8,9], which suggests potential mechanisms of how HMB might reduce the effects of IUGR and the incidence of low-birth-weight neonates. 

Researchers have studied dietary HMB supplementation in swine at various time points. Piglets born from sows supplemented with HMB were heavier at birth and had increased daily weight gain, allowing them to reach market weight faster than pigs farrowed by sows fed unsupplemented diets [10]. The dietary addition of HMB also enhanced protein synthesis in the skeletal muscle of newborn piglets [9]. The HMB-induced increase in the average birth weight of piglets previously observed [10] could result from an increase in the weight of all pigs or a decrease in the number of low-birth-weight pigs, resulting in improved uniformity of piglet birth weight. If the latter is true, HMB may mitigate the negative consequences of IUGR in pigs. However, few studies have examined HMB use in gestation diets to increase the uniformity of piglet birth weights within litters.

Unfortunately, sow studies are difficult to conduct due to the long time required for completion, high costs, extensive housing space requirements, and the large number of animals needed per treatment to detect differences among treatments, if they exist. Therefore, a suitable alternative reproductive model for sows is needed. Mice are a litter-bearing species that are also affected by IUGR [11]. We hypothesized that mice could serve as a reasonable model for the sow to evaluate the reproductive effects of dietary HMB on the birth weight variation and growth performance of offspring.

## 2. Materials and Methods

The experimental protocol used in this study was approved by the Institutional Animal Care and Use Committee of the University of Minnesota under protocol number 1709-35113A.

### 2.1. Animals and Experimental Design

This experiment was conducted in a Research Animal Resources Facility on the St. Paul campus of the University of Minnesota. Fifty-six post-pubertal, virgin, wild-type Agouti A^vy^ (93% C57bl/6 background, isogenic) dams [12] were used to conduct this experiment. Animals were randomly assigned to one of four dietary treatments: control (CON; *n* = 13), low-level HMB (LL; 3.5 mg/g; *n* = 14), high-level HMB (HL; 35 mg/g; *n* = 15), and pulse dosing of HMB (PUL; 3.5 mg/g; *n* = 14). Dams only received HMB during gestation. Dams assigned to the PUL treatment received the CON diet until day 6 of gestation, then switched to the LL diet until day 10 of gestation. They were then switched back to the CON diet for the remainder of gestation. Dams on the PUL treatment were assumed to be pregnant until pregnancy was confirmed on day 14. Days 6 to 10 of gestation were selected for the PUL treatment because the formation of the labyrinth structure of the placenta occurs during this time in mice [13]. The labyrinth structure is the location of nutrient transfer between the dam and fetus in mice. In each dietary treatment, half of the dams were euthanized on day 18 of gestation to collect each pup and its associated placenta. All dams were individually housed throughout the experiment in standard plastic cages (28 cm × 18 cm × 13 cm) with corn cob bedding and shredded paper for nesting material. Dams were provided ad libitum access to feed and water. During breeding, individual males were placed in cages with each female and were removed after 24 h of exposure. Pregnancy was confirmed 14 days after male exposure based on the increased body weight of dams. If dams were not pregnant, they were housed with a male again for another 24 h. The dam body weight was collected weekly throughout the 3-week gestation period and subsequent 4-week lactation period, and feed disappearance was recorded weekly for all dams. 

### 2.2. Dietary Treatments

Sires and dams received a nutritionally adequate diet based on corn and soybean meal before breeding. The diet was modeled after a sow lactation diet that is commonly used in the commercial swine industry. Diets (Table 1 and Appendix A) were blended offsite as a mash and subsequently pelleted in the laboratory. In brief, pellets were formed by first mixing 1 kg of the diet with 1600 mL of water to obtain a thick consistency. The diet-water mixture was then pelleted using the food grinder and sausage stuffer kit for a KitchenAid mixer (Pro 600™ Series 6 Quart Bowl-Lift Stand Mixer, Model No. KP26M1XER; KitchenAid, Benton Harbor, MI, USA) and placed on baking trays. Trays were then placed in a drying oven for approximately 12 h at 26–32 °C. Nutrient concentrations of the experimental diets compared with NRC nutrient requirements [14] for mice are shown in Appendix A. To make the experimental diets, β-hydroxy-β-methylbutyrate calcium salt (Ca-HMB; Hefei TNJ Chemical Industry Co., Ltd., Hefei, China) was mixed into the CON diet before pelleting, enabling the base of each diet to be the same. The concentration of HMB in the experimental diets was determined by Eurofins Microbiology Laboratories, Inc. (Des Moines, IA, USA) using the HPLC method for analysis.

At male exposure, diets fed to dams were switched to the experimental gestation diet that was randomly assigned. Dams designated for CON continued to receive the CON diet, but dams assigned to the LL and HL treatments were switched to a diet containing 3.5 or 35 mg HMB/g diet, respectively. Doses of HMB used in this experiment were extrapolated using allometric scaling by energy intake from the dose used in a previous sow study, as scaling by energy intake yields a more accurate dose than if calculated by body weight [15]. Briefly, scaling for the low dose was determined by calculating the amount of HMB sows consumed per kcal net energy in the study conducted by Tatara et al. [10]. For the LL treatment, 4.15 mg Ca-HMB (85% HMB) per gram diet was supplemented to provide 1.4 mg HMB/kcal net energy, which equaled the calculated dose used in a previous study [10]. Dams were provided ad libitum access to the CON diet throughout lactation. 

### 2.3. Birth Weight, Growth Performance, and Body Composition Measurements

The individual birth weight of pups and the total litter weight were recorded within 24 h of birth. Following birth, litters were weighed weekly. Body weight was also recorded for pups recovered from dams that were euthanized on day 18 of gestation (one day before expected parturition). 

At weaning, all pups were individually ear notched for identification and were sexed. All pups were weaned onto the CON diet and remained on this diet until 8 weeks of age. Weaned pups were caged by litter and sex. The individual body weight of pups and feed disappearance for each cage were recorded weekly. These data were used to calculate average daily gain (ADG), average daily feed intake (ADFI), and gain to feed (G:F).

At five and eight weeks of age, all pups were transferred to the University of Minnesota Phenotyping Core (Minneapolis, MN) for the measurement of body composition using magnetic resonance imaging (Echo MRI, Echo Medical System). Body weight was recorded at the time of scanning. The fat mass, lean mass, and total water of live pups were determined.

### 2.4. Histology

On day 18 of gestation, 28 dams were euthanized using CO_2_ inhalation to excise uterine horns and collect placental tissues and fetuses. Tissue samples were immediately placed in a 10% formalin solution for approximately 24 h. After 24 h, all tissue samples were rinsed with a PBS solution, transferred to a 70% ethanol solution, and stored at 4 °C until they were submitted for staining. Placental samples were embedded in paraffin using standard histological techniques, sectioned at a thickness of 4 µm, and stained with hematoxylin and eosin at the Comparative Pathology Shared Resource Laboratory (Masonic Cancer Center at the University of Minnesota, Minneapolis, MN, USA). Stained slides were evaluated by light microscopy using an Olympus BX53 Microscope (Center Valley, NJ, USA) at 4× power. CellSense imaging software (Version 1.18; Olympus, Center Valley, NJ, USA) was used to outline the total labyrinth area. The labyrinth area was determined in two separate tissue sections for each placenta. The average measurement of two sections was recorded as the area of the labyrinth. Measurements of 208 placentae were completed by one person who was blinded to treatments to reduce variation.

### 2.5. Gene Expression

Livers from offspring were flash-frozen at dissection. Three livers per treatment were randomly selected for RNA extraction using the RNeasy Mini Kit from Qiagen (Germantown, MD, USA) and the subsequent analysis of gene expression. A subsample (30 mg) was taken from each liver and placed in RNA*later* (Sigma-Aldrich; St. Louis, MO, USA) prior to homogenization and RNA extractions were completed following the manufacturer’s instructions. After extractions were completed, RNA was quantified in each sample using an Implen NanoPhotometer N50 (Implen, München, Germany) and diluted to 50 ng/µL. The analysis of gene expression in diluted samples was completed by the University of Minnesota Genomics Center (St. Paul, MN, USA).

RNA-sequencing (RNA-seq) was performed at the University of Minnesota Genomics Core, and RNA libraries were created using an 18 dual-indexed Illumina TruSeq Prep kit (Illumina, San Diego, CA, USA). All libraries were combined into a single pool and sequenced on an Illumina Nextseq instrument with a 1 × 76-bp run. This generated >35 M reads for each lane. All expected barcodes were detected and well represented, with mean quality scores of >Q30. Data were subsequently sent to the University of Minnesota Informatics Institute for genomics analysis. Library diversity indicated >35% reads remained after deduplication, with >97% of reads mapping to the mouse genome. The quality of data in fastq files was assessed using FastQC. Low-quality bases and adapter sequences were removed using Trimmomatic. Reads were aligned using Hisat2 FPKM expression values generated using Cuffquant and Cuffnorm from the Cufflinks package, and raw read counts were generated using featureCounts from the subread R package.

### 2.6. Statistical Analysis

For all analyses of animal performance, the GLIMMIX procedure of SAS (SAS Inst. Inc., Cary, NC, USA) was used. The breeding group was included in all statistical models as a random effect. The breeding group was defined as the contemporary group of dams that were successfully mated during a 2-week breeding period. Results are reported as least squares means, and comparisons among treatments were performed using the PDIFF option of SAS with the Tukey-Kramer adjustment for multiple comparisons. Treatment effects were considered significant if *p* < 0.05 and a trend if 0.1 > *p* ≥ 0.05.

Dam and litter performance data were analyzed as a completely randomized design where individual dams served as the experimental unit. The statistical model included dietary treatment as a fixed effect. To further evaluate birth weight, the two heaviest and two lightest birth weights per litter were assigned to either the high or low birth weight category, respectively. The birth weight, placental weight, placental efficiency (ratio of fetal weight:placental weight), and labyrinth area were analyzed using a statistical model that included the fixed effects of dietary treatment, birth weight category, and their interaction.

To investigate differences in variation of birth weight, weaning weight, and body weight of offspring at 8 weeks of age, standard deviations of each litter, ranges (difference between lightest and heaviest birth weight), and differences between median and lightest birth weights from each litter were calculated and compared among treatments. This analysis was repeated with litters that contained eight pups or more, which were considered to be large litters because they contained at least two more pups than the mean litter size for this mouse strain [16,17].

Repeated measures analysis was used to determine the effects of dam treatment during gestation on offspring performance and body composition from weaning to eight weeks of age. The statistical model for offspring performance and body composition included time (weeks), the dietary treatment of dams, sex, and all possible interactions of these three factors.

For gene expression data, the differential expression for each of six total pairwise comparisons (control vs. [LL, HL, PUL], low vs. [HL, PL], and HL vs. PUL) was determined based on a negative binomial generalized linear model with quasi-likelihood tests in the edgeR Bioconductor package (version 3.26.8). The *p*-values were adjusted using the Benjamini-Hochberg false discovery rate (FDR) approach with an FDR < 0.05 and two-fold cutoffs were used to determine significance. 

Gene enrichment analysis was performed using the ToppGene [18] suite matching the gene name to Ensemble ID. For enrichment analysis, all ribosomal genes were manually removed from the lists.

## 3. Results

### 3.1. Dam Performance in Gestation and Lactation

Dietary treatment had no effect on the overall gestation or lactation performance of mouse dams (Table 2). There was no effect of dietary HMB addition on body weight during the first two weeks of gestation; however, in the third week of gestation, the body weight of dams assigned to PUL tended to be greater (*p* < 0.10) than the body weight of dams assigned to HL. Dietary treatment had no effect on dam average daily gain (ADG) and average daily feed intake (ADFI). An interaction (*p* < 0.01) was observed between treatment and stage of gestation for ADFI of dams (Figure 1), indicating that there were no differences in ADFI among treatments during the first two weeks of gestation, but in the third week, dams assigned to HL had a lower (*p* < 0.01) ADFI than LL and PUL dams. A numerical decrease in ADFI was observed from week 2 to week 3 of gestation in dams assigned to HL, while all other dams had increased ADFI during this time period. During the 4-week lactation period, there were no differences among dietary treatments for body weight, ADG, and ADFI, and there were no significant interactions between dietary treatments and stage of lactation. 

### 3.2. Litter Performance and Placental Efficiency

There were no effects of dietary treatment during gestation on the total number of pups born alive per litter, the number weaned per litter, the average pup birth weight, the litter birth weight, or the litter weight at weaning (Appendix A). The dietary supplementation of HMB to dams during gestation had no effect on the placental weight or labyrinth area (Table 3). Placental efficiency, however, was lower (*p <* 0.05) in LL dams compared to CON dams, while dams assigned to PUL and CON treatments tended (*p* < 0.10) to have a higher placental efficiency than LL and HL dams. Dietary treatments did not influence the weight (birth weight and weight at gestational day 18), placental efficiency, or labyrinth area across birth weight categories (Table 4). Placental weights in the low-weight category tended to be higher (*p* < 0.10) in PUL dams compared with dams fed CON.

The variance in birth weight within litter and in the weaning weight among litters, expressed as either CV or SD, was not affected by the dietary treatment of dams during gestation (Appendix A). Ranges, which were calculated by the difference between the heaviest and lightest pup birth and weaning weights in each litter, were not different among treatments. However, at 8 weeks of age, the offspring of LL dams had a lower range (*p* < 0.05) in body weight per litter than the offspring of PUL dams. The difference between the median weight of each litter and the lightest pup of that litter, noted as the lower half, was similar among treatments at birth, weaning, and 8 weeks of age. Measurements of variance, including the within-litter CV for pup weight at birth and weaning, within-litter SD, range, and lower half of the range for birth weight, pup weaning weight, and body weight at 8 weeks of age in litters containing eight or more pups, were not influenced by dietary treatments (Appendix A).

### 3.3. Growth Performance of Offspring

Dietary treatments had no effect on the overall post weaning growth performance of offspring (Table 5). At 4 weeks post-weaning, offspring from LL dams had a greater (*p* < 0.05) individual body weight than offspring of HL dams. There was no effect of dietary treatments during the 4 weeks post weaning on ADG; however, there was a significant interaction (*p* = 0.02) between treatment and week post weaning. During the first week post weaning, offspring from PUL dams had greater ADG than those from all other treatments, but this advantage was lost during the second week when PUL offspring had a lower ADG than both CON and HL offspring. During the 4 weeks post weaning, dietary treatment tended to influence G:F. Offspring from HL and PUL dams displayed a greater (*p* < 0.05) G:F than LL offspring during the first week post weaning. However, during week two, G:F decreased (*p* < 0.05) for PUL offspring and was lower than that of HL offspring. There were no differences in G:F among treatments during weeks 3 or 4 post weaning (8 weeks of age).

### 3.4. Body Composition

The dietary treatments imposed on dams during gestation had small influences on the body composition of offspring (Figure 2). Offspring from LL dams were heavier (*p* < 0.05) at 5 weeks of age than offspring from CON and HL dams. Offspring from CON dams were also heavier (*p* < 0.05) than HL offspring. At the 8-week body scan, LL offspring continued to be heavier (*p* < 0.05) than those from all other dietary treatments. The lean mass percentage at 5 weeks of age was greater (*p* < 0.05) in HL offspring than in LL offspring. At 8 weeks of age, the lean mass percentage in HL offspring continued to be greater (*p* < 0.05) than in LL offspring. No differences were observed in fat mass at 5 weeks of age. However, at 8 weeks of age, offspring from the LL group had a higher (*p* < 0.05) fat mass percentage than offspring from PUL dams. The dietary treatment imposed on dams did not affect the total water percentage of offspring at 5 weeks of age. At 8 weeks, the offspring of PUL dams had a higher (*p* < 0.01) total water percentage than the offspring of LL dams. 

### 3.5. Gene Expression

Feeding diets containing β-hydroxy-β-methylbutyrate given to mouse dams altered gene expression in the livers of offspring (Table 6). In total, there were 60 genes in common that were affected by HMB supplementation when comparing dams fed CON with dams fed any other dietary treatment (Figure 3, Table 7). Space prohibits discussion of all 60 differentially expressed genes. Twenty-seven of these genes were classified as ribosomal-related genes, all of which play a role in RNA binding and with structural constituents of the ribosome. A few genes were downregulated, but one gene of particular interest was the gene that encodes for Ubiquitin (Ubb). 

A five-way list of overlapping genes indicated that the majority of genes differed between the control vs. LL groups, and the control vs. HL and PUL groups (Appendix A). For this reason, gene set enrichment analysis was performed using the 385 genes that were differentially expressed between the control group and the HL and PUL treatments combined. After removing 39 ribosomal genes, 346 genes were entered into the ToppGene suite. Of these, 295 genes were annotated in Ensembl. The gene ontology (GO) biological process assigned to these genes indicated strong enrichment in muscle-related functions (Appendix A). 

## 4. Discussion

The overall goal of this project was to evaluate β-hydroxy-β-methylbutyrate as a feed additive in practical sow diets to reduce IUGR. Applied experiments using breeding sows are very difficult to conduct because they require extensive replication due to large variations in the reproductive performance of sows caused by genetic and environmental variations typically present in sow herds. The large number of animals needed makes sow experiments very expensive and time-consuming to complete. In the experiment reported herein, we used mice as a model for sows because of their low cost, rapid reproductive rates, and low genetic variability. Mice and swine are both litter-bearing species with significant within-litter birth weight variation. A comparison of reproductive characteristics between mouse dams and sows is shown in Appendix A. The strain of mice used in this experiment had a within-litter birth weight coefficient of variation of about 10%, which is less than the 15% to 25% commonly observed in pig litters [11,19]. This difference in variation of body weight at birth between mice and sows is likely due to the fact that commercial swine genotypes are more genetically diverse than the mice used in this study and swine are subjected to more variable environmental factors than in mouse research colonies. The variation reported in the mouse strain used in this study is only the inherent variation because these mice were genetically identical and housed in a tightly controlled environment. Given that environmental and genetic sources of variation were dramatically smaller in mice than commercial sows, we expected to observe a birth weight response to HMB in mice even though the observed birth weight variation in mice is smaller than that observed in sows. 

Mice are susceptible to IUGR and variations in pup birth weights, even when they are genetically identical and housed under tightly controlled, constant environments. We acknowledge the structural differences between mouse and sow placentae, but the placentae are affected in a similar manner after GH/IGF-1 intervention. For both placental types, GH/IGF-1 increases the placental gene expression of Slc38a2, a gene for a neutral amino acid transporter in the tissue [20,21,22]. However, additional comparisons of placental function and nutrient transfer between the two species are needed to determine if the mouse is an acceptable model to study IUGR in sows. 

In the current study, the lack of dietary HMB effects on mouse dam performance during gestation and lactation was expected. Similarly, no adverse effects on sow performance during gestation, lactation, or subsequent reproductive cycles have been reported when supplementing HMB in sows during late gestation and lactation [23]. In the third week of gestation, we noted a decrease in the feed intake of dams in the HL group. High dietary HMB inclusion may have created an aversion to the diet that caused the decline in feed intake. However, if an aversion to a high concentration of HMB existed, we would have expected to see a decline in feed intake when the diets were first introduced and not after 2 weeks of consumption. Baxter et al. fed rats 5% dietary HMB for 90 days and reported that a 5% inclusion in the diet resulted in no observed adverse effects [24]. Therefore, we did not expect that the HL dose at 4.3% of the diet would decrease feed intake. However, the rats used in the Baxter et al. toxicology study [24] were not pregnant or lactating. As a result, there may have been an interaction between the high concentration of dietary HMB and pregnancy in the present experiment that resulted in the decreased feed intake. Furthermore, the body weights of mouse dams were not different among dietary treatments until the third week of gestation, when dams fed HL had a lower body weight compared to dams assigned to the other three treatments. This reduction in body weight was most likely caused by the reduction in feed intake observed for HL dams during this same period.

The birth weight of pups was not affected by maternal HMB supplementation during gestation. These findings agree with previous research conducted in swine [23,25,26]. However, other studies have reported an increase in birth weight from maternal diet HMB supplementation in pigs [10,27,28,29]. Inconsistent results of the effects of dietary HMB addition on birth weight may be due to the varying lengths of feeding HMB-supplemented diets. Swine studies with positive results have been reported when dietary HMB supplementation began two weeks before farrowing [10,27] or as early as day 35 of gestation through farrowing [28]. In previous swine experiments where no differences in birth weight were observed, dietary HMB supplementation was provided only 3 to 10 days before farrowing [23,25], which suggests that supplementation occurred too late in gestation for any effect on the birth weight to be observed. In the current study, supplementation was either constant throughout the entire gestation period or was only provided during a critical time in placental development. Therefore, we expected to observe a positive effect of maternal diet HMB supplementation on pup weight at the end of gestation. 

Another potential reason for the variable responses to dietary HMB supplementation in previous sow experiments may have been the wide range of HMB doses fed. In the studies where dietary HMB supplementation had no effect on birth weight, sows consumed 2 to 3 g HMB/day, equating to 0.08 to 0.14% of the total diet or about 9.8 mg HMB per kg body weight. This is considerably lower than the low dose used in the current study, which was about 1000 mg HMB per kg of dam body weight. In swine studies showing increased birth weight or increased blood-borne growth hormone, sows were fed 50 mg Ca-HMB per kg body weight in the last 2 weeks of pregnancy [10,27]. In studies using Ca-HMB, the concentrations of HMB in the Ca-HMB products used were not reported, making the exact amount of HMB supplemented difficult to determine. Concentrations of Ca-HMB products range from 80% HMB to 99% HMB. With the Ca-HMB product used in the current study containing 85% HMB, the LL dose provided 0.35% HMB in the diet. Without knowing the actual concentration of HMB in the Ca-HMB products used in studies that demonstrated positive effects on piglet birth weight, one cannot accurately determine the most effective HMB dose. 

The supplementation of dam diets with HMB had no effect on the within-litter variation of birth weight based on the standard deviation of birth weights and the coefficient of variation within litters. This was the most important and unique response variable measured in this experiment. Very few researchers have reported the effects of HMB on within-litter variation in terms of birth weights in favor of reporting the average birth weight of entire litters. Of the researchers that have measured this trait, Tatara et al. found no effect of dietary HMB on within-litter birth weight variation for neonatal pigs [27]. In contrast, supplementing HMB from day 35 of gestation through parturition reduced the percentage of piglets born weighing less than 1 kg compared to unsupplemented sows [28]. Supplemental β-hydroxy-β-methylbutyrate can increase endogenous levels of growth hormone [10]. When providing sows with exogenous growth hormone, fetal and placental weights increased [30]. Growth hormone can selectively improve the environment for smaller pigs in utero by influencing placental nutrient transfer and placental growth [31]. β-hydroxy-β-methylbutyrate can increase the activity of the GH/IGF-1 axis [10,27,32]. If growth hormone selectively improves growth conditions for smaller piglets in utero, then this may be a potential mechanism to reduce the incidence of low-birth-weight pigs from sows fed diets supplemented with HMB. Our data presented herein suggest HMB is not effective in reducing the occurrence of low-birth-weight progeny in a mouse model, but we did not measure the circulating growth hormone in mouse dams. In future studies, concentrations of growth hormone in dams and their offspring should be measured to determine the effects of dietary HMB. 

Researchers do not know if HMB can be transferred across the placenta to fetuses. However, if HMB does not cross the placenta to directly act on the fetus, HMB may still have an indirect effect by increasing growth hormone production in the dam and influencing placental nutrient transfer and placental growth [31]. In the current study, we did not find any differences in placental development with HMB supplementation. Future research may explain any potential effects of HMB on placental function, such as increasing placental nutrient transporters. 

The daily pattern of HMB intake may influence its effect on the birth weight of progeny. Hu et al. [33] studied the absorption and metabolism of HMB fed to pregnant sows once daily. They reported that the net portal flux of HMB peaked at 30 min after a meal and approached zero at about 6 h post prandial. They suggested HMB be consumed multiple times daily to maintain elevated plasma concentrations of HMB. Mouse dams in the present study were allowed ad libitum access to dietary treatments. However, the majority of voluntary feed intake (70%) in mice occurs at night [34]. This diurnal pattern of HMB consumption may have been insufficient to maintain continuously high plasma concentrations of HMB in mouse dams and, therefore, limited the efficacy of HMB in our experiment. 

The weaning weights of offspring were not different among treatments in this study. Previously, researchers have noted that pigs from dams fed diets supplemented with HMB during late gestation through lactation had heavier weaning weights, even if there were no differences in birth weights [23]. The increase in weaning weight with dietary HMB supplementation may be a result of an increase in milk fat percentage in sows supplemented with HMB during lactation [23,25,28]. The increase in milk fat could have increased the energy intake of offspring, leading to an increased growth rate compared with offspring from unsupplemented sows that had a lower milk fat percentage. Mouse offspring in the present study may not have had different weaning weights because all dams were fed the same basal diet during lactation with no HMB supplemented. 

No differences in the overall growth performance measures of offspring among treatments were observed. In the present study, the body weight of mice at 8 weeks of age was greatest in the LL group but was not different among the other dietary treatments. This response is different from previous studies in livestock species such as pigs [35], cattle [36], and broiler chickens [37], where growth parameters such as ADG, ADFI, or G:F were increased with HMB supplementation. In some of these studies, the test animals were fed HMB during the post-weaning growth period [36,37]. In our study, the offspring did not receive HMB after birth and, therefore, this may be a reason why we did not observe the same results in our experiment.

One of the mechanisms of action of HMB is to increase protein synthesis and reduce proteolysis [9,38,39,40,41]. The body composition of mouse offspring was not altered by maternal HMB supplementation in this experiment. In other species, HMB supplementation increased the lean meat percentage at harvest [10,35,37]. Less abdominal fat and increased marbling were also observed [36,37]. In all of these previous studies, apart from Tatara et al. [10], animals received HMB supplementation during the postnatal growth period. Mice in the current study were not fed HMB after birth, which may explain why no effects on body composition were observed. 

Previous researchers noted the increased expression of a gene that corresponds with the GH/IGF-1 axis when feeding HMB to rats [32]. Unfortunately, we were not able to replicate that outcome in the present trial. However, we determined, through our gene expression and pathway analysis, that HMB affects the expression of genes in muscle-related pathways. One example of this is the downregulation of the *Ubb* gene. Ubiquitin is known for its major role in targeting cellular proteins for degradation, and because it was downregulated, it coincides with a known mode of action of HMB, which is the reduction of proteolysis [6,7,8,9]. This finding helps provide more evidence about HMB’s mode of action and how it is effective in preventing muscle breakdown. The magnitude of change in gene expression was within the expected range. Similarly, the higher number of gene expression changes in the HL treatment compared with other treatments was expected. There is a commonality between HMB treatments unrelated to the dose or duration of supplementation. Gene ontology of the common genes among HMB treatments was heavily related to membrane transport, which suggests another area that requires further study.

## 5. Conclusions

The goal of the research presented herein was to determine if HMB could reduce within-litter birth weight variation and improve growth performance in mice. Mice were chosen to serve as an economical surrogate model for sows. We found that supplementing diets of mouse dams with HMB during gestation had no effects on the birth weight of pups, within-litter birth weight variation, placental weight, labyrinth area, pup weaning weight, or growth performance and body composition of offspring after weaning. The daily frequency of HMB feeding and the utility of the mouse model as a surrogate for sows requires further investigation. 

## Figures and Tables

**Figure 1 animals-13-03227-f001:**
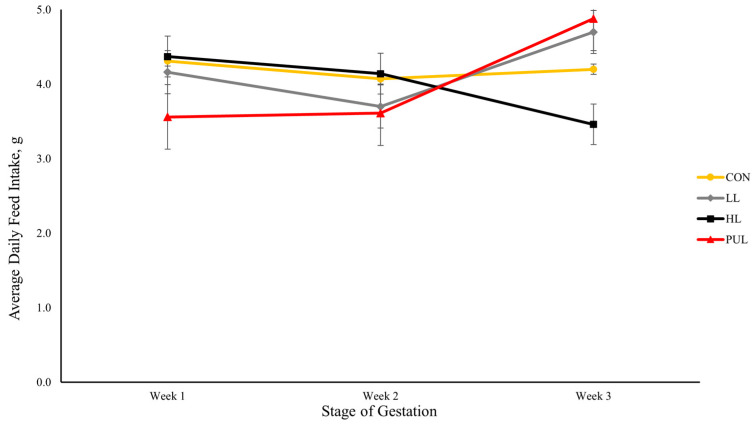
Interaction of dietary treatments and stage of gestation for average daily feed intake of dams. CON = Control; LL = Low-level HMB supplementation; HL = High-level HMB supplementation; PUL = Pulse dose of low-level HMB during days 6 to 10 of gestation.

**Figure 2 animals-13-03227-f002:**
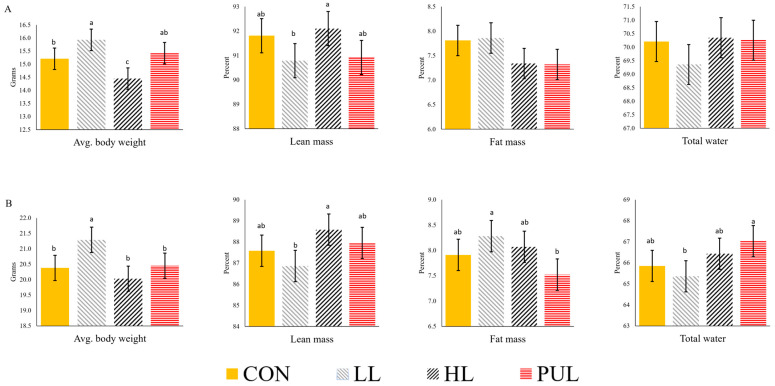
Effects of maternal HMB supplementation during gestation on the body composition of offspring. Data shown at 5 weeks of age (**A**) and 8 weeks of age (**B**). CON = Control; LL = Low-level HMB supplementation; HL = High-level HMB supplementation; PUL = Pulse dose of low-level HMB during days 6 to 10 of gestation. ^a,b,c^ Bars with different superscripts are different (*p* < 0.05) within each response variable.

**Figure 3 animals-13-03227-f003:**
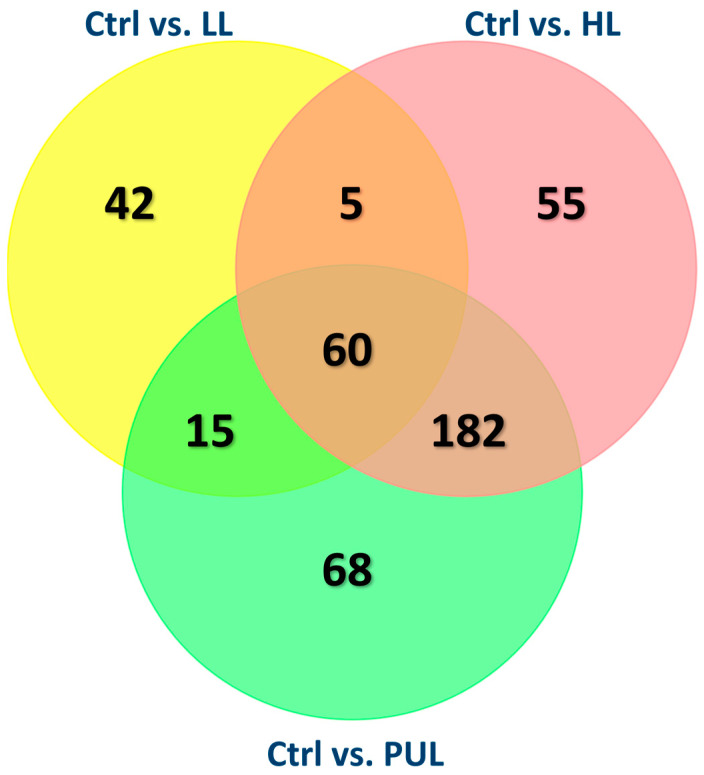
Venn diagram depicting the number of altered genes that overlapped among treatments.

**Table 1 animals-13-03227-t001:** Diet composition, as fed %.

Ingredient	Diet ^1^		
	CON	LL	HL
Corn, ground	63.18	62.91	60.46
Soybean meal, 46.5% Crude protein	29.30	29.17	28.04
Mineral mix ^2^	3.50	3.48	3.35
Vitamin mix ^3^	1.00	1.00	0.96
Calcium carbonate	0.27	0.27	0.26
Choline chloride	0.20	0.20	0.19
Soy oil	2.55	2.54	2.44
Ca-HMB ^4^, 85% HMB	0.00	0.43	4.30
Total	100.00	100.00	100.00

^1^ CON = Control; LL = Low-level HMB supplementation; HL = High-level HMB supplementation. ^2^ Contained the following ingredients per kg of premix: calcium carbonate, 357 g; potassium phosphate, monobasic, 196 g; potassium citrate, H_2_O, 70.78 g; sodium chloride, 74 g; potassium sulfate, 46.6 g; magnesium oxide, 24 g; ferric citrate, U. S. P., 6.06 g; zinc carbonate, 1.65 g; manganous carbonate, 0.63 g; cupric carbonate, 0.3 g; potassium iodate, 0.01 g; sodium selenite, 10.25 mg; ammonium paramolybdate tetrahydrate, 7.95 mg; sodium metasilicate, 0.9 H_2_O, 1.45 g; chromium potassium sulfate dodecahydrate, 0.275 g; lithium chloride, 17.40 mg; boric acid, 81.50 mg; sodium fluoride, 63.50 mg; nickel carbonate, 31.80 mg; ammonium vanadate, 6.60 mg. ^3^ Contained the following ingredients per kg of premix: niacin, 3.0 g; calcium pantothenate, 1.60 g; pyridoxine HCl, 0.70 g; thiamine HCl, 0.60 g; riboflavin, 0.60 g; folic acid, 0.20 g; biotin, 0.02 g; vitamin E acetate, 7500 IU; vitamin B12, 0.1%, 2.50 g; vitamin A palmitate, 400,000 IU; vitamin D3, 100,000 IU; vitamin K1, 75 mg. ^4^ β-Hydroxy-β-methylbutyrate calcium salt; Sourced from Hefei TNJ Chemical Industry Co., Ltd., Hefei, China. HMB was mixed with the control diet before pelleting.

**Table 2 animals-13-03227-t002:** Effect of dietary treatment on the gestation and lactation performance of dams.

	Treatments ^1^					*p* Value	
Trait	CON	LL	HL	PUL	SE	Trt ^2^	Trt × Time ^3^
No. of dams	13	14	15	14	-	-	
Gestation:							
Avg. body weight, g					1.32	0.01	0.60
Week 1	21.07	20.53	20.44	21.05			
Week 3	32.03 ^xy^	30.99 ^xy^	29.27 ^x^	33.66 ^y^			
Avg. daily gain, g	0.25	0.28	0.16	0.27	0.05	0.21	-
Avg. daily feed intake, g	4.40	4.27	4.12	4.38	0.28	0.76	-
Lactation ^4,5^:							
Avg. body weight, g					1.33	0.12	0.66
Week 1	27.10	28.12	25.23	27.75			
Week 4	28.56	29.04	27.97	27.52			
Avg. daily gain, g	0.05	0.06	0.12	0.09	0.04	0.61	-
Avg. daily feed intake, g	9.09	9.15	8.98	10.34	0.74	0.27	-

^1^ CON = Control; LL = Low-level HMB supplementation; HL = High-level HMB supplementation; PUL = Pulse dose of low-level HMB during days 6 to 10 of gestation. ^2^ Dietary treatment. ^3^ Interaction between dietary treatment by stage of gestation or lactation. ^4^ Number of dams per treatment used in analysis: CON, *n* = 6; LL, *n* = 7; HL, *n* = 8; PUL, *n* = 7. ^5^ 28-day lactation period. ^xy^ Means within a row with different superscripts differ (*p* < 0.10).

**Table 3 animals-13-03227-t003:** Effect of dietary treatments on placental characteristics.

	Treatments ^1^					
Trait	CON	LL	HL	PUL	SE	*p* Value
No. of litters	7	6	7	7	-	-
No. of pups	59	43	48	58	-	-
Placental weight, g	0.24	0.24	0.24	0.25	0.008	0.47
Placental efficiency ^2^	4.415 ^a,x^	3.967 ^b,y^	3.971 ^ab,y^	4.353 ^ab,x^	0.20	< 0.01
Labyrinth area, mm^2^	5.10	5.17	5.06	5.03	0.16	0.89

^1^ CON = Control; LL = Low-level HMB supplementation; HL = High-level HMB supplementation; PUL = Pulse dose of low-level HMB during days 6 to 10 of gestation. ^2^ Placental efficiency = (fetal weight/placental weight). ^ab^ Means within a row with different superscripts differ (*p* < 0.05). ^xy^ Means within a row with different superscripts differ (*p* < 0.10).

**Table 4 animals-13-03227-t004:** Effect of dietary treatment on fetal weight and placental characteristics across birth weight categories ^1^.

	Treatments ^2^					*p* Value	
Trait	CON	LL	HL	PUL	SE	Trt ^3^	Trt × WtCat ^4^
No. of litters	13	14	15	14	-	-	-
Pup weight ^5,6^, g					0.07	0.11	0.98
High	1.28	1.28	1.22	1.32			
Low	1.11	1.07	1.03	1.12			
Placental weight ^7^, g					0.01	0.17	0.13
High	0.259	0.256	0.249	0.258			
Low	0.222 ^y^	0.233 ^xy^	0.234 ^xy^	0.257 ^x^			
Placental efficiency ^7,8^					0.35	0.22	0.62
High	4.40	4.03	4.19	4.50			
Low	4.32	3.86	3.80	3.90			
Labyrinth area ^7^, mm^2^					0.25	0.65	0.37
High	5.21	4.92	4.95	4.98			
Low	4.87	4.85	5.21	4.87			

^1^ Birth weight categories were determined by selecting the two heaviest and two lightest weight pups within each litter for the High and Low categories, respectively. ^2^ CON = Control; LL = Low-level HMB supplementation; HL = High-level HMB supplementation; PUL = Pulse dose of low-level HMB during days 6–10 of gestation. ^3^ Dietary treatment. ^4^ Interaction of dietary treatment and birth weight category. ^5^ Contains birth weights and fetal weights from pups euthanized at gestational day 18. ^6^ No. of litters: CON = 13; LL = 14; HL = 15; PUL = 14. ^7^ No. of litters: CON = 7; LL = 6; HL = 7; PUL = 7. ^8^ Placental efficiency = (fetal weight/placental weight). ^xy^ Means within a row with different superscripts differ (*p* < 0.10).

**Table 5 animals-13-03227-t005:** Effect of dietary treatment fed to dams on post-weaning performance of offspring.

	Treatments ^1^					*p* Value
Trait	CON	LL	HL	PUL	SE	Trt ^2^
No. of litters	6	8	8	7	-	-
No. of mice	40	55	61	53	-	-
No. of cages	13	17	20	17	-	-
Avg. body weight, g					0.50	<0.01
Weaning	10.87 ^ab^	11.37 ^a^	11.27 ^a^	10.16 ^b^		
Final	20.65 ^ab^	20.91 ^a^	19.73 ^b^	20.24 ^ab^		
Avg. daily gain, g	0.34	0.31	0.32	0.34	0.02	0.12
Avg. daily feed intake, g	2.75	2.73	2.53	2.81	0.17	0.24
G:F ^3^	0.12	0.12	0.12	0.12	0.01	0.54

^1^ CON = Control; LL = Low-level HMB supplementation; HL = High-level HMB supplementation; PUL = Pulse dose of low-level HMB during days 6–10 of gestation. ^2^ Dietary treatment of dam. ^3^ Gain to feed ratio. ^ab^ Means within a row with different superscripts differ (*p* < 0.05).

**Table 6 animals-13-03227-t006:** Total number of genes with two-fold changes between treatments.

	Treatment Comparisons ^1^					
Trait	CON vs. LL	CON vs. HL	CON vs. PUL	LL vs. HL	LL vs. PUL	HL vs. PUL
Total	122	302	325	254	248	0
Up regulated	112	90	108	1	1	0
Down regulated	10	212	217	253	247	0

^1^ CON = Control; LL = Low-level HMB supplementation; HL = High-level HMB supplementation; PUL = Pulse dose of low-level HMB during days 6–10 of gestation.

**Table 7 animals-13-03227-t007:** List of common genes with fold changes from all HMB treatments compared with CON.

Gene ^2^	Treatments ^1^			*p*-Value ^3^
	CON vs. LL	CON vs. HL	CON vs. PUL	
*Ass1*	22.34	23.44	23.84	<0.05
*Atp5g2*	3.38	3.36	3.24	<0.01
*BC100530*	3.28	3.25	3.24	<0.05
*Bnip3*	4.60	2.72	3.51	<0.05
*Btf3l4*	3.04	2.55	2.97	<0.05
*Cks2*	3.57	3.37	3.87	<0.01
*Cox5b*	2.48	2.29	2.59	<0.05
*Dnajc19*	2.61	2.55	2.86	<0.05
*Eef1a1*	2.97	2.44	2.68	<0.05
*Erh*	5.92	4.98	5.20	<0.01
*Fabp5*	2.97	2.57	3.01	<0.01
*Gm13315*	−21.71	−32.58	−16.91	<0.01
*Gm5124*	−12.88	−13.39	−23.13	<0.05
*Gm6548*	3.22	3.12	3.35	<0.01
*H3f3a*	7.31	5.99	8.40	<0.01
*Hadhb*	2.25	2.54	2.25	<0.05
*Hmgn2*	3.94	4.34	3.99	<0.01
*Hspa8*	2.28	2.58	3.03	<0.05
*Lsm7*	4.22	3.39	4.62	<0.05
*Mif*	2.77	2.80	2.54	<0.05
*Mir682*	22.94	20.07	17.92	<0.01
*Mpc1*	3.31	3.04	2.91	<0.05
*Mrps36*	3.66	3.5	3.25	<0.05
*Nsa2*	4.76	3.71	4.22	<0.01
*Paox*	3.27	3.45	3.44	<0.05
*Pgk1*	2.72	2.17	2.45	<0.05
*Ppia*	397.39	370.22	492.11	<0.01
*Psenen*	3.85	4.39	3.00	<0.05
*Stfa2*	2.86	3.32	2.38	<0.05
*Sumo2*	8.27	5.68	6.08	<0.01
*Tpt1*	5.18	4.25	5.53	<0.01
*Ubb*	−4.34	−4.18	−6.43	<0.01
*Uqcrb*	3.55	3.65	3.66	<0.01

^1^ CON = Control; LL = Low-level HMB supplementation; HL = High-level HMB supplementation; PUL = Pulse dose of low-level HMB during days 6–10 of gestation. ^2^ Ribosomal-related genes have been excluded from this table. ^3^ *p* values for individual comparisons all fall under reported value.

## Data Availability

The raw data for RNA sequencing is available at the National Institutes of Health SRA website under accession number PRJNA893690.

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
