# Peer review of "Evaluation of Feeding Beta-Hydroxy-Beta-Methylbutyrate (HMB) to Mouse Dams during Gestation on Birth Weight and Growth Variation of Offspring"

_animals, 2023, doi:10.3390/ani13203227_

Round 1

Reviewer 1 Report

The experiments are well designed and the description is well done. Methods are well explained. The discussion is well argued.

For embryonal mice studies, I always suggest cervical dislocation as euthanasia cause CO2 could affeced the tissue analysis.

In my opinion the paper is suitable for pubblication.

Author Response

The experiments are well designed and the description is well done. Methods are well explained. The discussion is well argued.

            Response:  Thank you.

For embryonal mice studies, I always suggest cervical dislocation as euthanasia cause CO2 could affeced the tissue analysis.

            Response:  Thank you for your suggestion.  We will consider that for future experiments.  Obviously, we cannot change our approach now. 

In my opinion the paper is suitable for pubblication.

            Response:  Thank you.

Reviewer 2 Report

In view of similar problem in the swine (with quite serious economic consequences) the suggestion of improving the homogeneity of the litter by adding anything or treating the pregnant dam, the dietary supplementation with HMB is an interesting approach. I understand that after some former swine experiments the authors wanted to establish a similar but cheaper mouse model to study whether the nutritional supplementation with HMB during pregnancy (or a part of pregnancy) could help the low body weight at birth. However, the question arises why only one essential amino acid (leucin) has been checked. And also, if the much lower birth weight variation in the mouse (around 10%) allows conclusions that can be extrapolated to the pig with a double birth weight variation. It is not the failure of the study that this hypothesis did not decrease the birth weight variation, just like in former pig experiments. This article analyses the issue quite profoundly and found that (in spite of optimized dosages) HMB supplementation could not solve the issue. Some explanations are given in the conclusions and the hypothesis of down regulation of Ubiquitin (Ubb) gene is quite interesting, worth of continuation. For the reviewer the question remains open why one treated group (PUL) showed larger difference compared to the two other treatment groups (LL and HL).

Author Response

In view of similar problem in the swine (with quite serious economic consequences) the suggestion of improving the homogeneity of the litter by adding anything or treating the pregnant dam, the dietary supplementation with HMB is an interesting approach. I understand that after some former swine experiments the authors wanted to establish a similar but cheaper mouse model to study whether the nutritional supplementation with HMB during pregnancy (or a part of pregnancy) could help the low body weight at birth. However, the question arises why only one essential amino acid (leucin) has been checked.

            Response:  We selected HMB (a leucine metabolite) for this study based on previous work reported by Tatara et al. (2007 & 2012) in which supplementation of pregnant sows with HMB improved birthweight of piglets.  We selected only the leucine metabolite to be consistent with Tatara’s work.  If we had supplemented multiple essential amino acids, we would not know which amino acid or amino acids were responsible for any responses observed. 

 And also, if the much lower birth weight variation in the mouse (around 10%) allows conclusions that can be extrapolated to the pig with a double birth weight variation. It is not the failure of the study that this hypothesis did not decrease the birth weight variation, just like in former pig experiments.

Response:  Good point.  Our research team discussed this at some length as we designed the experiment.  We decided that the variation in birth weight present in this line of mice is reasonably comparable to commercial sows when one considers that the mouse dams were genetically identical, were housed in an environmentally-controlled laboratory, and fed a constant diet.  We decided that the birth weight variation observed in the mice when genetic and environmental variation was almost non-existent would be similar to the birth weight variation seen in commercial sows when genetics, nutrition, and housing environment are responsible for a much greater portion of the phenotypic variation in birth weight observed on farms.  We added text in L404-408 to explain this rationale for the reader.

This article analyses the issue quite profoundly and found that (in spite of optimized dosages) HMB supplementation could not solve the issue. Some explanations are given in the conclusions and the hypothesis of down regulation of Ubiquitin (Ubb) gene is quite interesting, worth of continuation. For the reviewer the question remains open why one treated group (PUL) showed larger difference compared to the two other treatment groups (LL and HL).

Response:  Yes, we similarly wondered if this larger difference for PUL has biological significance.  We do not have a plausible explanation for this apparently larger magnitude of difference. 

Reviewer 3 Report

The authors describe a study on feeding HMB to mouse dams during gestation. This was done against the background of high organizational and financial demands of respective studies in pigs. Consequently, mice dams were used as surrogates for sows. The study is well designed and all methods are described sufficiently detailed. However, there are some questions to be answered and some modifications of the manuscript are necessary. Major points to be addressed are:

(I) The originality of this study is somewhat limited as there exist several studies on supplementation of sows during gestation. This raises the question why a surrogate model is necessary. A brief look into the literature reveals that even studies on HMB metabolism in sows during gestation and on transfer of HMB from sows to their offspring were published (e.g. Wu et al. 2017 – cited; Hu et al. 2020 – not cited here). Therefore, the authors should elaborate on the specific, unanswered question to be addressed in the mouse model.

(II) Why was this specific mouse line used? Are there features of this line making it a suited swine model?

(III) The manuscript is by far too long. It is acknowledged that negative results should be published. However, this manuscript details negative results on stunning 28 pages with 11 tables and 6 figures. Figures 2-4 may be transferred into a supplement (or even be deleted) and the same is true for several tables dealing with the diets. There are several redundant paragraphs in the manuscript (e.g. but not limited to l80-83 vs. l442-447).

(IV) The discussion on gene expression is too narrow. The authors focus on a single gene (Ubb) out of 60 DEGs.

(IV) The conclusion is a summary of the results and largely repeats the statements given in the last paragraph of the discussion. On the contrary, I miss a clear statement regarding the limited value of this specific mouse model for swine.

Minor:

- l407-410 is discussion not results

- Appendix is no standard feature in this journal. Tables and figures (see also [III] should be published as a separate supplement.

- Bibliography of references 3, 19, 22 is wrong or incomplete

Author Response

The authors describe a study on feeding HMB to mouse dams during gestation. This was done against the background of high organizational and financial demands of respective studies in pigs. Consequently, mice dams were used as surrogates for sows. The study is well designed and all methods are described sufficiently detailed. However, there are some questions to be answered and some modifications of the manuscript are necessary. Major points to be addressed are:

(I) The originality of this study is somewhat limited as there exist several studies on supplementation of sows during gestation. This raises the question why a surrogate model is necessary. A brief look into the literature reveals that even studies on HMB metabolism in sows during gestation and on transfer of HMB from sows to their offspring were published (e.g. Wu et al. 2017 – cited; Hu et al. 2020 – not cited here). Therefore, the authors should elaborate on the specific, unanswered question to be addressed in the mouse model.

            Response:  Good question.  There are several issues wrapped up in responding to this question. 

1.  Yes, there are swine studies published in which HMB was fed to sows.  In all but one of those studies, birth weight responses were measured on a litter basis such that individual piglet birth weight and the variation in piglet birth weight within litter was not reported.  This is the important difference for our study where we were most interested in the within litter birth weight variation and the incidence of IUGR offspring.  Existing text and text we added in this revision hopefully brings this point to greater prominence (L18, 396-408, 470-490, 565-569).

2.  We added a short discussion of the Hu et al. study (L502-511) as suggested. This study sheds some light on why literature reports are inconsistent with regard to HMB’s influence on birth weight of progeny.

3.  There is some gray area around whether HMB is an approved feed additive for pigs in the U.S.  As far as we know, HMB is not approved by the American Association of Feed Control Officials (AAFCO) for inclusion in swine diets.  Therefore, the mouse model has some utility since these animals will not enter the food chain.

(II) Why was this specific mouse line used? Are there features of this line making it a suited swine model?

            Response:  Dams in this mouse line are as close to genetically identical as one can get.  So, using these mice should control for genetic variation present in commercial sows.  The mice in this study were housed in a tightly-controlled environmental room with consistent temperature, air exchanges, and lighting.  This tight environmental control provided a consistent environment for conduct of the experiment, an environment much more consistent than could be provided in any sow housing accommodations.  Consequently, our model controlled genetic and environmental variables to a high degree allowing us to study directly the effects of dietary HMB. 

(III) The manuscript is by far too long. It is acknowledged that negative results should be published. However, this manuscript details negative results on stunning 28 pages with 11 tables and 6 figures. Figures 2-4 may be transferred into a supplement (or even be deleted) and the same is true for several tables dealing with the diets. There are several redundant paragraphs in the manuscript (e.g. but not limited to l80-83 vs. l442-447).

            Response:  Thanks for the comment.  We struggled with how much data to present.  And, we were concerned about the length of the manuscript.  We have moved Tables 2, 9, 10, 11, A1 and Figures 2, 3, 4, A1 to Supplementary Materials.  This should streamline the paper tremendously.

(IV) The discussion on gene expression is too narrow. The authors focus on a single gene (Ubb) out of 60 DEGs.

            Response:  We selected one gene to discuss as an example because of its relevance to our knowledge of HMB actions described in previous literature.  Obviously, we cannot discuss all 60 genes and we realize this reviewer is not asking us to do so.  However, how many genes is enough?  2, 5, 10?  We chose one gene especially given the concern with unduly increasing the length of the manuscript. 

(IV) The conclusion is a summary of the results and largely repeats the statements given in the last paragraph of the discussion. On the contrary, I miss a clear statement regarding the limited value of this specific mouse model for swine.

            Response:  We have deleted the summary paragraph (L558-565) and slightly modified the Conclusions section (L571-573).  We included a statement in the conclusions raising questions about use of the mouse model as suggested (L574). 

Minor:

- l407-410 is discussion not results

            Response:  Agreed.  We moved this text to L547 in the Discussion section.

- Appendix is no standard feature in this journal. Tables and figures (see also [III] should be published as a separate supplement.

            Response:  A collection of Supplementary Materials has been developed and submitted with this revision.

- Bibliography of references 3, 19, 22 is wrong or incomplete

            Response:  Citations have been corrected.  Format for citation 22 varies from Animals requirement because Biology of Reproduction includes an Article number with the volume and page numbers.

Round 2

Reviewer 3 Report

My comments have been addressed adequately. The ms. was shortened and is now acceptable for me.